# Investigation of Viscoelastic Properties of Polymer-Modified Asphalt at Low Temperature Based on Gray Relational Analysis

**Zhongcai Huang [1], Rong Lu [2],\*, Zhiyu Fu [2], Jingxiao Li [2], Pengfei Li [2], Di Wang [3],\*, Ben Wei [1], Weining Zhu [1], Zujian Wang [1] and Xinyu Wang [1]**

1    Guangxi Communications Investment Group Corporation Ltd., Nanning 530022, China; hzc17377103066@163.com (Z.H.)
2    School of Highway, Chang'an University, Xi'an 710064, China
3    Department of Civil Engineering, Aalto University, 02150 Espoo, Finland
*    Correspondence: lurong@chd.edu.cn (R.L.); di.wang@aalto.fi (D.W.)

**Abstract:** As the investigation indexes of low-temperature viscoelastic properties of polymer-modified asphalt (PMA) are unclear at present, in this paper, the creep stiffness ($S$), creep rate ($m$), low-temperature continuous classification temperature ($T_C$), $\Delta T_C$, $m/S$, relaxation time ($\lambda$), and dissipation energy ratio ($W_d(t)/W_s(t)$) were taken as a comparison sequence. The maximum flexural tensile strain ($\varepsilon_B$) of porous asphalt mixture (PAM) in a low-temperature bending test was selected as a reference sequence. Gray relational analysis was used to investigate the PMA's low-temperature viscoelastic properties based on a bending beam rheometer (BBR). The results show certain contradictions in investigating the low-temperature properties of PMA when only considering the low-temperature deformation capacity or the stress relaxation capacity. The modulus and relaxation capacity should be considered when selecting the investigation indexes of the low-temperature viscoelastic properties of PMA. When rheological method is used to evaluate the low-temperature of polymer modified asphalt, $T_C$ and $m/S$ are preferred. When only $S$ or $m$ is contradictory, $m$ should be preferred. $\Delta T_C$ can determine whether the low-temperature performance of PMA is dominated by $S$ or $m$. The result can better guide the construction of asphalt pavement in areas with low temperatures. Asphalt can be selected quickly and accurately to avoid the waste of resources.

**Keywords:** road engineering; polymer-modified asphalt; gray relational analysis; low temperature; viscoelastic characteristics; evaluation index

## 1. Introduction

Porous asphalt pavement has been widely concerned because of its excellent properties such as permeability, noise-reduction and skid-resistance [1–3]. Polymer-modified asphalt significantly improves porous asphalt pavement performance and meets the demand for high-quality asphalt [4–6]. It is used to make asphalt binders by adding polymer modifiers, such as rubber, polymer, and other admixtures, into the matrix asphalt [7,8]. Under low-temperature conditions, the main deteriorations of polymer-modified asphalt pavement are low-temperature cracking, loose, and pit. Low-temperature cracking (LTC) is a particularly common disease in asphalt pavement [9–11]. It is mostly due to the low winter temperature and the road surface temperature decrease. After that, the stiffness modulus of the asphalt material becomes high, and the shrinkage phenomenon occurs [12–14]. It could affect the integrity, continuity, and driving comfort of pavement [15]. Moreover, it can lead to the softening of the subgrade, the decline of pavement bearing capacity, and the acceleration of the destruction of asphalt pavement [16,17]. Studies have shown that LTC in asphalt pavement is highly associated with the performance of asphalt binders at low temperature [18,19]. Ma et al. [20] studied LTC in northeast China, and the results showed that increasing the fine aggregate content could extend the anti-cracking performance of asphalt pavement at low temperatures. Li et al. [21] established the asphalt pavement's

mechanical model to analyze the causes of pavement cracking. They combined with a ductility test, asphalt composition analysis test, and bending beam rheometer (BBR) to investigate the behavior of LTC under thermal and light conditions. The experimental results indicated that the anti-cracking ability of asphalt at low temperature degrades under the influence of heat and light. The composition of the asphalt has a critical effect on its performance at low temperature [10]. SBS modifier can help control the LTC of asphalt [22]. In order to effectively improve the pavement service function, it is necessary to study the low-temperature performance of asphalt binders and determine the effective low-temperature performance evaluation index. Researchers have put forward many evaluation indexes, mainly low-temperature elongation, low-temperature penetration, creep stiffness, creep rate, performance grade (PG) grading, dissipated energy ratio, relaxation time, and $m/S$ [23–29]. Zhang et al. [30] investigated the cold behavior of TB rubber, SBS-modified asphalt, and its mixture by trabecular BBR and a semicircular bending stretch test (SCB). The findings revealed that the PG grading of TB rubber-modified asphalt could well reflect the asphalt performance in the low-temperature range. In contrast, the low-temperature performance of asphalt mixture incorporating TB rubber with SBS-modified asphalt cannot be as well evaluated by a single PG grading, so it needs to be assessed together with other indexes. Zhou et al. [31] discussed the cold behavior of SBS and CR-modified asphalt based on the BBR test at three different temperatures. From the perspective of energy, the energy dissipation ratio and damping coefficient of asphalt were calculated to evaluate the ability of asphalt in low-temperature environments. The analysis showed that adding CR to SBS-modified asphalt was beneficial to reducing the LTC and improving the two factors above, before, and after the aging process. As far as the present research is concerned, it is better to use a simple fractional-order viscoelastic model for creep analysis of asphalt at low temperature. In order to study the effects of asphalt relaxation characteristics at low temperature, Zheng et al. [32] carried out beam bending relaxation tests at low temperature by using the dynamic thermomechanical analyzer. Relaxation rate and relaxation time are two indexes that can best indicate the relaxation characteristics of asphalt at present. The results show that these two indexes can effectively characterize the performance of asphalt at low temperature. Bai et al. [24] studied the low-temperature properties of a SBS-modified asphalt binder. The results show the statistical linear correlation between the different low-temperature indices. However, penetration below 0 °C may not be a valid index to evaluate the low-temperature performance of asphalt binders. Wang et al. [33] used a curved-beam rheometer to measure the creep stiffness modulus of six types of ordinary asphalt and six types of modified asphalt under different temperatures and loading times based on the time-temperature equivalence principle. The results show that the stiffness index has a clear physical meaning and can better reflect the low-temperature performance of asphalt compared with the conventional index.

In addition, Sun et al. [34] compared the evaluation indexes of the rubber asphalt performance at low temperatures and demonstrated that the $m/S$ was preferred in engineering. The evaluation index $S$, which comprehensively considered the modulus and relaxation capacity of asphalt, was selected in scientific research. Dong et al. [35], found that the creep rate greatly affected the low-temperature performance and used the m/S to evaluate the low-temperature performance of SBS-modified asphalt. Wei et al. [36] studied the polyphosphoric acid-modified asphalt performance at low temperatures and found that integrated flexibility was the optimal evaluation index for PPA-modified asphalt at low-temperature. Wu et al. [37] used the creep stiffness and creep rate as the indexes to compare PE, SBS, and SBR-modified asphalts by BBR test. Yan et al. [38] studied the conventional properties and low-temperature rheological properties of four types of PMA and found that the glassy transition temperature had the highest correlation, which could better reflect the low-temperature properties of PMA, followed by the low-temperature rheological indexes and viscosity indexes. Gu et al. [39] studied the evaluation indexes of the low-temperature performance of foamed warm mixed crumb rubber-modified asphalt and indicated that the low-temperature ductility index is not suitable. At the same



time, the glass transition temperature is feasible as the evaluation index for evaluating the low-temperature performance of rubber-modified asphalt. Huang et al. [40] studied the evaluation indexes of the low-temperature performance of SBS-modified asphalt. The results showed that low-temperature elongation was not suitable for evaluating the low-temperature performance of SBS-modified asphalt. Xu et al. [41] studied the evaluation indexes of the low-temperature performance of the warm-mix-modified asphalt. The results showed that the low continuous grading temperature could more accurately evaluate the low-temperature performance of the warm-mix-modified asphalt. Geng et al. [42] studied the evaluation indexes of low-temperature properties of high-modulus asphalt. The results showed that the creep stiffness could not accurately evaluate the low-temperature proper­ties, so it was suggested to use the fracture energy of the single-notch curved-beam test as the evaluation index of high modulus asphalt. Gao et al. [43] conducted research on low temperature damage by analyzing cooling conditions and long-term aging. The cooling rate and creep rate decay index are proposed to predict the service life of actual pavement at low temperature. In summary, the evaluation index of low-temperature viscoelastic properties obtained from the bending creep stiffness test can accurately reflect the cold behavior of asphalt binder. Still, there is no consistent conclusion on which evaluation index to use. The reliability of the evaluation index of asphalt binders must be judged com­prehensively based on the properties of the asphalt mixture. This study aimed to explore the evaluation index of low-temperature viscoelastic properties of PMA. First, we selected the PMA (SBS, rubber, and high-viscosity-modified asphalt) and seven evaluation indexes of low-temperature viscoelastic properties, which include $S$ (creep stiffness), $m$ (creep rate), $T_C$ (low-temperature continuous classification temperature), $\Delta T_C$, $m/S$, $\lambda$ (relaxation time) as well as $W_d(t)/W_s(t)$ (dissipated energy ratio). Then, we tested the maximum flexural tensile strain ($\varepsilon_B$) of PAM in a low-temperature bending test. After that, we used gray relational analysis to clarify the evaluation indexes of the viscoelastic properties of PMA at low temperatures. Finally, we recommended the evaluation index of PMA's viscoelastic properties.

## 2. Material and Methods

### 2.1. Materials

Several common and popular asphalts were selected in this study. Styrene-butadiene-styrene-modified asphalt (SBS) and crumb rubber-modified asphalt (CRMA) is all finished modified asphalt. High-viscosity-modified asphalt (HVMA-I, HVMA-II) was prepared using SK-90 matrix asphalt and two high-viscosity modifiers (named Type I and Type II in this paper). The content of the high-viscosity modifier was 12% of the mass of matrix asphalt. The process of HVMA-I and HVMA-II is as follows: The asphalt high-speed shear was first conducted at 2000 r/min for 10 min and then at 5000 r/min for 30 min under 170–180 °C. It was kept for 10 min at 175 °C after the shear was completed. The fundamental properties of the bitumen mentioned above are shown in Table 1.

**Table 1.** Technical properties of asphalt.

| Type | Penetration (25 °C, 5 s, 100 g)/(0.1 mm) | Softening Point/°C | Ductility (5 °C)/cm | Dynamic Viscosity (60 °C)/(Pa·s) | Perfomance Grade |
|---|---|---|---|---|---|
| SK-90 | 97.1 | 47.4 | 9.7 | 140.3 | / |
| HVMA-I | 51.1 | 84.3 | 64.3 | 38,696.9 | PG64-22 |
| HVMA-II | 54.6 | 85.9 | 59.9 | 20,425.1 | PG64-22 |
| SBS | 64.0 | 94.2 | 45.7 | 14,169.2 | PG58-22 |
| CRMA | 51.1 | 63.2 | 13.3 | 3177.7 | PG52-22 |

The gradation of PAC-13 (the nominal maximum particle size of porous asphalt concrete is 13 mm) is shown in Table 2. The coarse aggregate, fine aggregate, and mineral

powder used diabase, limestone, and limestone mineral powder, respectively. The fiber was lignin fiber (the dosage is 3‰ of aggregate mass).

**Table 2.** Gradation of PAC-13.

| Sieve Size/mm | 16 | 13.2 | 9.5 | 4.75 | 2.36 | 1.18 | 0.6 | 0.3 | 0.15 | 0.075 |
|---|---|---|---|---|---|---|---|---|---|---|
| **Passing Ratio (by pass)/%** | 100.0 | 92.3 | 70.4 | 18.9 | 15.9 | 13.0 | 11.1 | 8.2 | 6.7 | 4.6 |

### 2.2. Test Methods

The bending beam rheometer (BBR) test of PMA was made according to JTG E20-2011, as shown in Figure 1a,b. The sizes of specimens are 127 mm long, 6.35 mm thick, and 12.70 mm wide. The load of BBR is 980 mN and the loading time is 240 s. This study's selected test temperature was $-12\,°C$, $-18\,°C$, and $-24\,°C$.

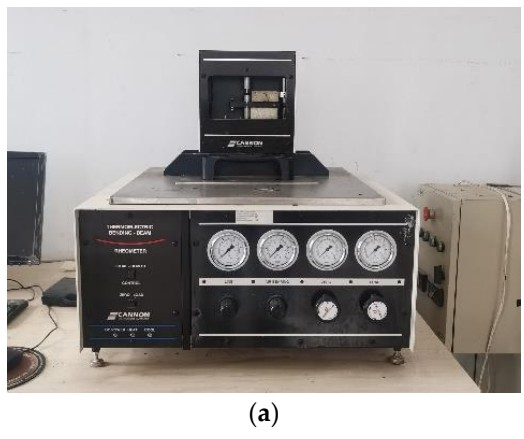

(**a**)

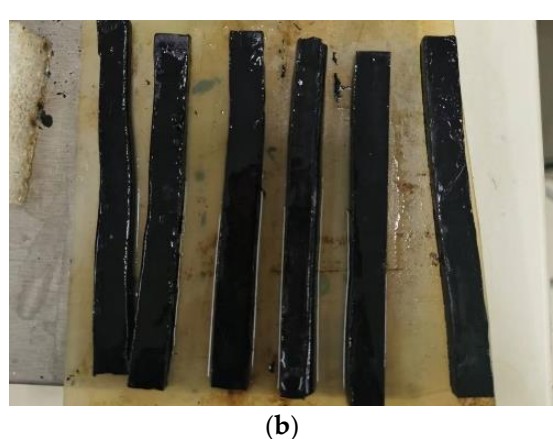

(**b**)

**Figure 1.** BBR and PMA's specimens. (**a**) BBR, (**b**) Specimens of PMA.

The low-temperature bending test was designed to evaluate the cold behavior of PAM. The test temperature was $-10\,°C$, the loading rate was 50 mm/min, the trabecular size was 250 mm in length, 30 mm in width, and 35 mm in height, and the span was 200 mm. The electronic universal testing machine of Mester Industrial System (China) Co., Ltd. (Zhongshan, China) was used. The test machine and specimens are as shown in Figure 2a,b.

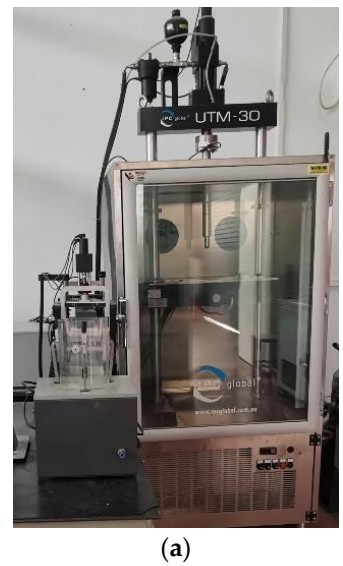

(**a**)

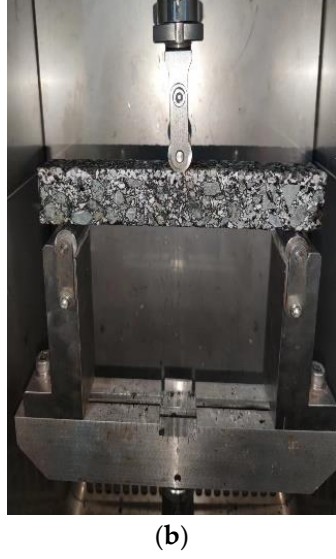

(**b**)

**Figure 2.** (**a**) Universal testing machine, (**b**) Specimens of low-temperature bending test.

*2.3. Flowchart*

In this study, five kinds of polymer-modified asphalt were selected, and the $S$, $m$, $T_C$, $\Delta T_C$, $m/S$, $\lambda$ as well as $W_d(t)/W_s(t)$ were obtained and calculated by BBR test, and the $\varepsilon_B$ were obtained by low-temperature bending test. Finally, the best indexes for evaluating the low-temperature performance of asphalt mixture were selected by gray relational analysis. The flowchart is shown in Figure 3.

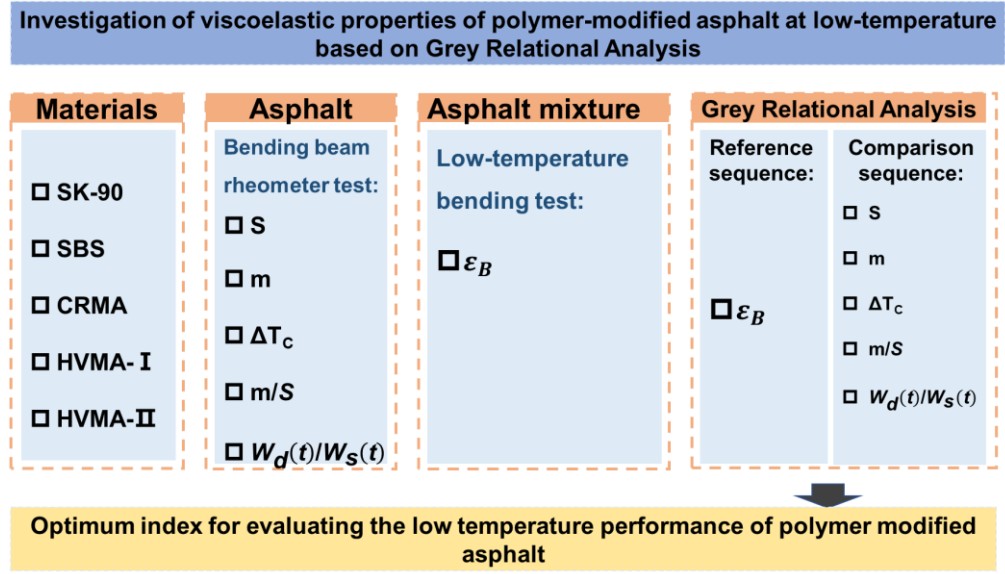

**Figure 3.** Flowchart of the research.

## 3. Results and Discussions

*3.1. Creep Stiffness and Creep Rate*

Generally speaking, the lower the creep stiffness ($S$), the higher the creep rate ($m$), indicating the better the performance of asphalt at low temperature [44,45]. The $S$ and $m$ values of the four PMAs are shown in Figures 4 and 5. As temperature drops, the $S$ of asphalt increases, the $m$ decreases, and the resistance to LTC is poor. $S$ and $m$ of PMA have no obvious rules under different low-temperature conditions. Moreover, $S$ and $m$ are not synchronized in evaluating modified asphalt under the same temperature conditions. SHRP (Strategic Highway Research Program) prescribed $S \leq 300$ MPa of BBR test for 60 s, and $m \geq 0.30$ were used to identify the cold behavior of asphalt materials. As can be observed in Figures 4 and 5, when the test temperature was $-24\,°C$, the $S$ and $m$ of PMA exceeded SHRP program requirements. However, the $S$ and $m$ of four PMAs met the recommended values of the specification at $-18\,°C$. Thus, the evaluation index of BBR at $-18\,°C$ was chosen to investigate the low-temperature performance of PMA. It can be seen that the $S$ value of SBS was the highest at $-24\,°C$, followed by CRMA, HVMA-I, and HVMA-II. The $S$ value of the four PMA at $-18\,°C$ was in the order of HVMA-I, CRMA, SBS, and HVMA-II. HVMA-II had a better cracking resistance at low temperature than HVMA-I, CRMA, and SBS at $-18\,°C$ for $S$. The trend of $S$ at $-12\,°C$ was similar to that at $-18\,°C$, but the value of CRMA decreased to a greater extent than that of the other three PMAs. In the four PMAs, the m values were CRMA, SBS, HVMA-II, and HVMA-I in descending order, and the trend was the same at the three temperatures. That is, CRMA has better low-temperature performance than HVMA-I, HVMA-II, and SBS. Using $S$ and $m$ to evaluate PMA for low-temperature performance is contradictory.

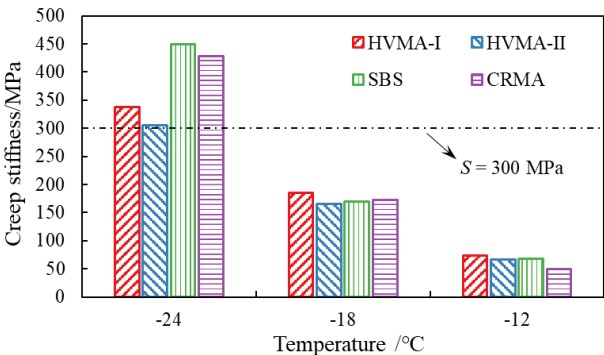

**Figure 4.** Creep stiffness of four different asphalts.

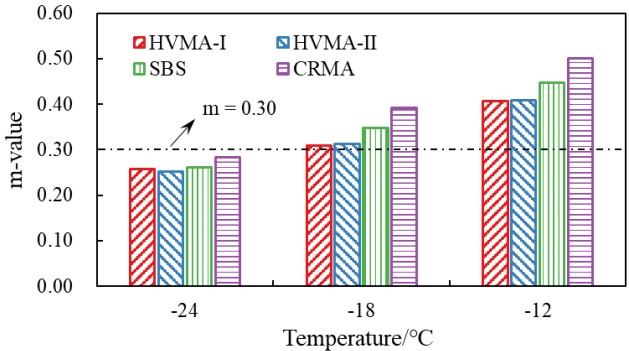

**Figure 5.** *m*-value of four different asphalts.

### 3.2. Low-Temperature Continuous Classification Temperature

SHRP proposed using performance grade (PG) to evaluate asphalt performance, which has been widely used because of its practicability and effectiveness. However, some studies have shown some limitations in using PG to evaluate asphalt low-temperature performance. That is, PG can only distinguish the asphalt low-temperature performance under different classification temperatures but cannot effectively assess the asphalt performance at low temperature at the same classification temperature [46]. Thus, ASTM D 7643-10 interpolates with $S = 300$ Mpa and $m = 0.30$ to calculate the fractional temperature difference between $S$ and $m$, and ultimately uses the greater of the two as the low-temperature continuous fractional temperature ($T_C$) [47]. The calculation method of $T_C$ is shown in Formulas (1) and (2). The $T_C$ of the four PMAs is shown in Figure 6. It can be noted that the $T_C$ from large to small are HVMA-I, HVMA-II, SBS, and CRMA. The calculated values of SBS and CRMA showed little difference, while HVMA-I and HVMA-II were slightly larger than those of the above two PMAs, indicating that CRMA has the lowest low-temperature continuous classification temperature and good low-temperature performance.

$$T_C = T_1 + \left( \frac{\log_{10}(P_S) - \log_{10}(P_1)}{\log_{10}(P_2) - \log_{10}(P_1)} \right)(T_2 - T_1) \tag{1}$$

where $T_C$ is the low-temperature continuous classification temperature, °C; $T_1$ and $T_2$ are the calculated temperature, °C, and $T_2$ is 6 °C higher than $T_1$; $P_S = 300$; $P_1$ and $P_2$ are the m corresponding to $T_1$ and $T_2$.

$$T_C = T_1 + \left( \frac{P_S - P_1}{P_2 - P_1} \right)(T_2 - T_1) \tag{2}$$

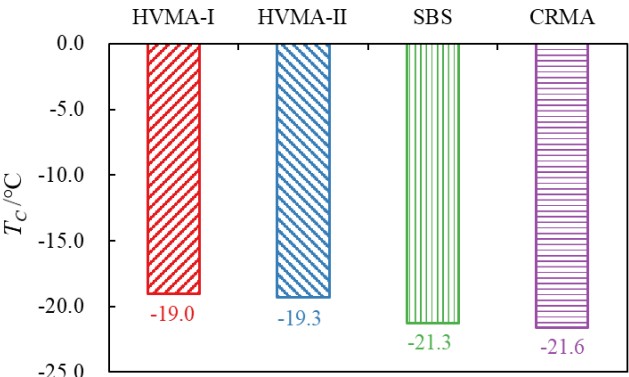

**Figure 6.** Low-temperature continuous grade temperature of asphalt.

### 3.3. $\Delta T_C$

$\Delta T_C$ represents the balance degree between the stiffness and the stress relief capacity of asphalt under low-temperature conditions [48]. The positive values of $\Delta T_C$ indicate that the asphalt performance at low temperature is mainly controlled by $S$, and the negative values of $\Delta T_C$ indicate that the asphalt low-temperature performance is primarily controlled by $m$. The absolute value of $\Delta T_C$ suggests the degree of asphalt controlled by $S$ or $m$. $S = 300$ Mpa and $m = 0.30$ are used to calculate $\Delta T_C$ by the interpolation method, and it is similar to the low-temperature continuous classification temperature. The calculation principle is illustrated in Figure 7, and the specific calculation method is given in Formulas (3)–(5). It can be found in Figure 8 that the low-temperature performance of CRMA is mainly controlled by $S$. The low-temperature performance of HVMA-I, HVMA-II, and SBS is primarily influenced by $m$. In addition, the stress release ability of HVMA-II at low temperatures is weaker than that of HVMA-I and SBS.

$$T_{c,S} = T_1 + \left( \frac{(T_1 - T_2) * (\text{Log } 300 - \text{Log } S_1)}{\text{Log } S_1 - \text{Log } S_2} \right) - 10 \tag{3}$$

$$T_{c,m} = T_1 + \left( \frac{(T_1 - T_2) * (0.300 - m_1)}{m_1 - m_2} \right) - 10 \tag{4}$$

where $S_1$ is the creep stiffness at time $t_1$, Mpa; $S_2$ is the creep stiffness at $t_2$, MPa; $m_1$ is the creep rate at time $T_1$; $m_2$ is the creep rate at $T_2$; $T_1$ and $T_2$ are the calculated temperatures respectively, °C, and $T_2$ is higher than $T_1$.

$$\Delta T_C = T_{c,S} - T_{c,m} \tag{5}$$

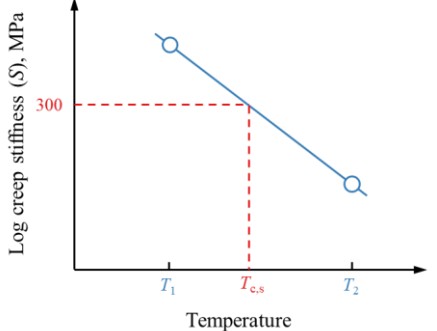
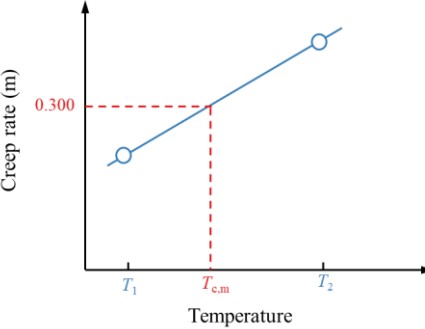

**Figure 7.** Graphical concept of $Tc,s$ and $Tc,m$.

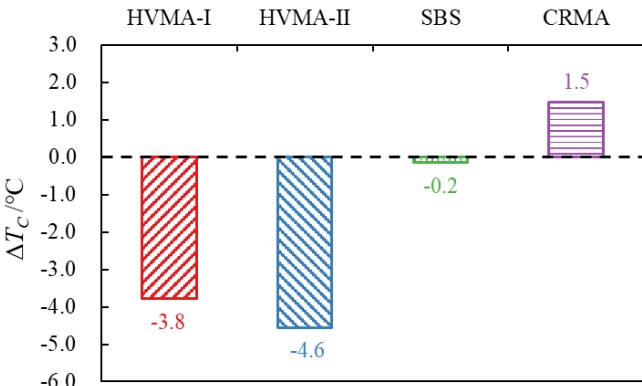

**Figure 8.** $\Delta T_C$ of different asphalt.

### 3.4. m/S

Relevant studies show that the ratio of *m* to *S* can also be used to characterize asphalt performance at low temperature [49]. The higher the *m/S*, the asphalt materials exhibit better low-temperature performance. Figure 9 shows the *m/S* of the four PAMs at different low temperatures. The *m/S* of the PMA were arranged from largest to smallest is CRMA, SBS, HVMA-II, and HVMA-I, respectively. It indicates that CRMA performs well at low-temperature performance, and the ranking of the *m/S* is consistent with that of low-temperature continuous grading and m. The increasing rate of the *m/S* of the four asphalts in the temperature range from −24 °C to −18 °C is less than that from −18 °C to −12 °C. It is because as the temperature rises, the energy of the molecular movement within the asphalt increases, and the movement of the molecular chain segments in the asphalt becomes more active. Thus, it can lead to the activity of the molecular structure of the asphalt.

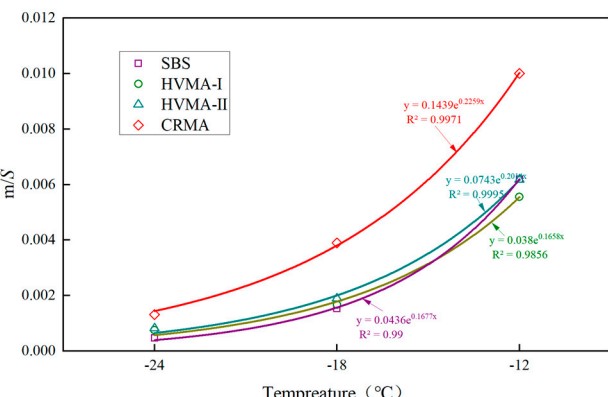

**Figure 9.** *m/S* value of different asphalt.

### 3.5. Burgers Model

PMA is a typical viscoelastic material, and its properties can be effectively described using the Burgers model, a four-element viscoelastic constitutive model. As shown in Figure 10, the Burgers model was obtained by combining the Kelvin model and the Maxwell model. Its mathematical formula is as follows [50]:

$$\varepsilon(t) = \sigma_0 \left[ \frac{1}{E_1} + \frac{1}{\eta_1}t + \frac{1}{E_2}\left(1 - e^{-\frac{E_2}{\eta_2}t}\right)\right] \tag{6}$$

where $\varepsilon$ is strain; $\sigma_0$ is the stress applied, MPa; $E_1$ is instantaneous elastic modulus; $\eta_1$ is instantaneous viscosity coefficient; $E_2$ is the slow deformation that occurs after the stress is applied; $\eta_2$ is the viscosity index that the deformation does not disappear immediately after removing the applied stress.

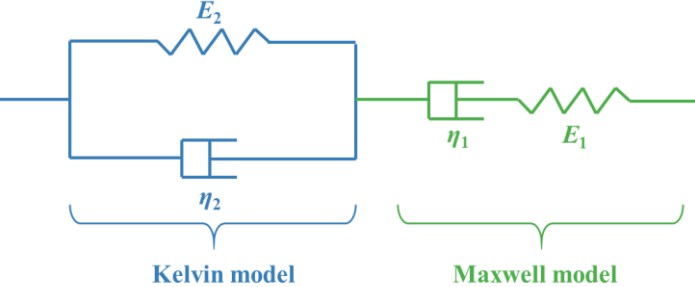

**Figure 10.** The Burgers model.

Based on the Burgers model, the BBR data of the four PMAs were nonlinearly fitted by 1stOpt-First Optimization software. The fitting parameters $E_1$, $E_2$, $\eta_1$, $\eta_2$ were used to calculate the relaxation time $\lambda$ and dissipated energy ratio ($W_d(t)/W_s(t)$), calculated by the following formula [16]:

$$\lambda = \eta_1/E_1 \tag{7}$$

$$W_d(t)/W_S(t) = \left[\frac{t}{\eta_1} + \frac{1}{2E_2}\left(1 - e^{-\frac{2E_2}{\eta_2}t}\right)\right] \Big/ \left[\frac{1}{E_1} + \frac{1}{2E_2}\left(1 - 2e^{-\frac{E_2}{\eta_2}t} + e^{-\frac{2E_2}{\eta_2}t}\right)\right] \tag{8}$$

where $t$ is the stress action time, s.

Relaxation time reflects the measurement of the stress variation of the asphalt binder with time. The longer the relaxation time, the more adverse it is to the rapid dissipation of the stress of the asphalt binder [51]. Formula (7) is used to calculate the relaxation time of the asphalt, as shown in Figure 11. It indicates the relaxation time of four types of asphalt from large to small: HVMA-I, HVMA-II, SBS, and CRMA in the temperature change process. CRMA exhibits the shortest relaxation time and performs best at low temperature. The relaxation time of the asphalt will increase with the decrease in the temperature, because the elasticity of the asphalt increases, and the viscosity decreases as the temperature drops. Further, the energy consumption rate slows while the stress change takes longer. Finally, the relaxation time becomes longer.

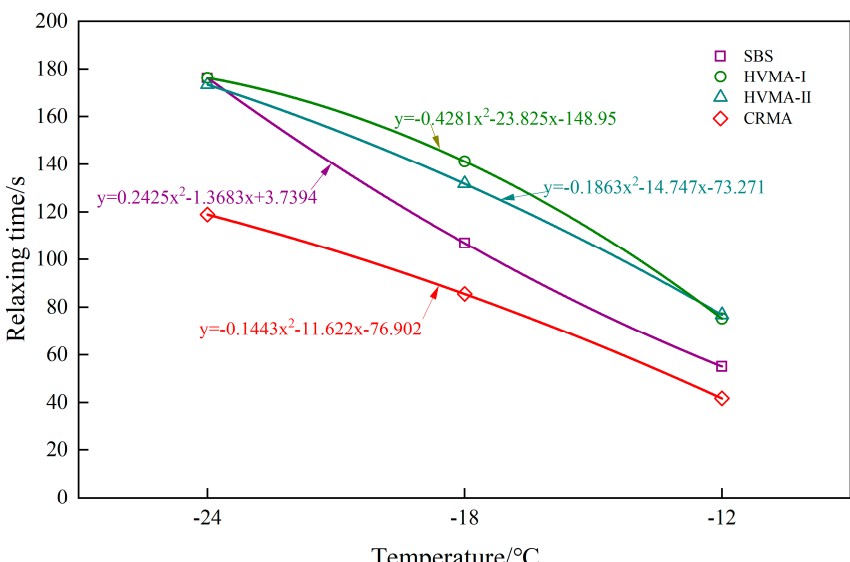

**Figure 11.** The relaxation time of asphalts under different temperatures.

The dissipated energy ratio is a parameter reflecting the relaxation capacity of the asphalt binder. The greater the dissipated energy ratio, the greater the LTC resistance of the asphalt binder [17]. Formula (8) was used to calculate the dissipative energy ratio. It can be

viewed from Figure 12 that the dissipative energy ratio of the four PMAs is in the order of CRMA, SBS, HVMA-II, and HVMA-I in the process of temperature change. It indicates that CRMA has the best performance at low temperature, in accordance with the evaluation result of the relaxation time. The dissipation energy ratio of the asphalt decreases with the lower temperature, indicating that the dissipation energy in the asphalt decreases with the decrease in temperature. Moreover, it increased the storage energy and poor resistance to LTC. The dissipated energy ratio of the four bitumen drops rapidly in the temperature range from −18 °C to −12 °C. The dissipated energy ratio of the four asphalts declines slightly when the temperature range is −24 °C to −18 °C. It is indicated that the elastic ratio of the bitumen increases obviously with the decrease in the temperature, and the bitumen is close to an elastic body when the temperature drops to a certain degree.

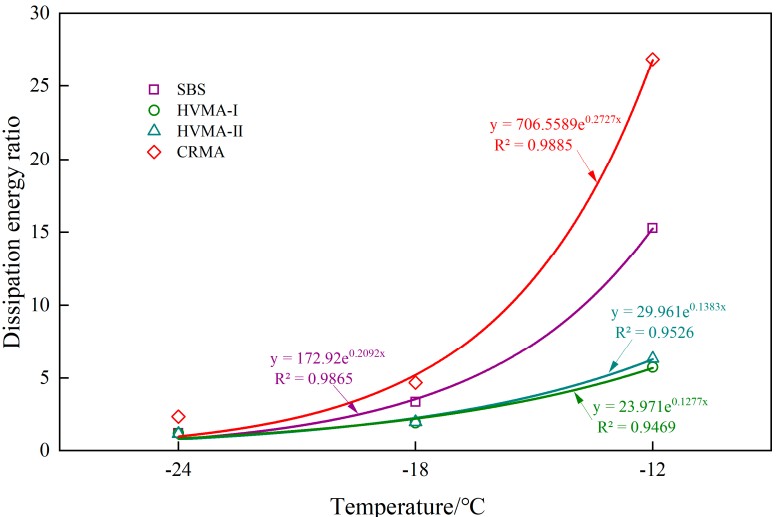

**Figure 12.** The dissipation energy ratio of asphalt under different temperatures.

### 3.6. Low-Temperature Bending Test of PAM

In this study, the trabecular specimens of porous asphalt mixture made from four kinds of polymer-modified asphalt were used to test the low-temperature performance of the mixture. The $\varepsilon_B$ is the maximum flexural tensile strain of a rectangular beam, which is applied to illustrate the PAM performance at low temperature during the specimen failure of the low-temperature bending test. $\varepsilon_B$ represents the ultimate deformation resistance of the asphalt mixture at low temperature. Its mathematical formula is shown in Formula (9).

$$\varepsilon_B = \frac{6 \times h \times d}{L^2} \tag{9}$$

where $L$ is the trabecular span, mm; $h$ is the trabecular height, mm; $d$ is the deflection of trabeculae during failure, mm.

As shown in Table 3, the LTC resistance increases as the $\varepsilon_B$ value rises. Further, the LTC resistance of the PAM formed by the four kinds of PMAs is in the order of CRMA, SBS, HVMA-II, and HVMA-I. That is, CRMA has the best LTC resistance, and HVMA-I has the worst LTC resistance.

**Table 3.** Results of PAC-13's low-temperature bending beam test.

| Type | HVMA-I | HVMA-II | SBS | CRMA |
|------|--------|---------|-----|------|
| $\varepsilon_B / \mu \varepsilon$ | 3281.5 | 3498.8 | 3547.9 | 3956.1 |

### 3.7. Gray Relational Analysis

Gray relational analysis is mainly based on the geometric shape of the sequence curve to judge the similarity between two sequences. Usually, the linear interpolation method is adopted to convert the observation data of discrete behavior of the observation system into piecewise continuous lines, and then the corresponding model is constructed according to the geometric characteristics of the lines. Then, the corresponding model is used to judge the similarity between the sequences [52]. Gray relational analysis can analyze engineering systems under "small samples and poor information" and determine the primary and secondary system influencers [53,54]. The basic step is to calculate the gray correlation coefficient (Formula (10)) and then calculate the gray correlation entropy (Formula (11)) and gray entropy correlation (Formula (12)). The higher the gray relational analysis of the comparison column, the stronger the correlation between the comparison and the reference column. The connection between the comparison column and the reference column can be reflected by its gray correlation degree. A higher correlation degree indicates a higher correlation.

$$\xi_i[x_0(k), x_i(k)] = \left| \frac{\min\limits_{i=1,m} \min\limits_{k=1,n} \Delta_i(k) + \rho \max\limits_{i=1,m} \max\limits_{k=1,n} \Delta_i(k)}{\Delta_i(k) + \rho \max\limits_{i=1,m} \max\limits_{k=1,n} \Delta_i(k)} \right| \tag{10}$$

where $\rho$ is the resolution coefficient and is 0.5; $x_0[x_0(1), \cdots, x_0(n)]$ is a reference sequence; $x_i[x_i(1), \cdots, x_i(n)]$ $(i = 1, 2, \ldots, m)$ is the comparison sequence.

$$H(R_i) \triangleq -\sum_{k=1}^{n} P_h \ln P_k \tag{11}$$

where $R_i = \{\xi[x_0(k), x_i(k)] | k = 1, \cdots, n\}$; $P_h \triangleq \frac{\xi[x_0(h), x_i(h)]}{\sum_{k=1}^{n} \xi[x_0(h), x_i(h)]}$, $P_h \in P_i (h = 1, \cdots, n)$.

$$E(x_i) \triangleq H(R_i)/H_{max} \tag{12}$$

where $H_{max} = lnn$, which represents the maximum value of n. Based on the previous analysis, this study selected evaluation indexes of $S$, $m$, $T_C$, $\Delta T_C$, $m/S$, $\lambda$, and $W_d(t)/W_s(t)$ for PMAs at $-18\ ^\circ$C as a comparison sequence and of the low-temperature bending test as a reference sequence to conduct gray entropy analysis. Table 4 shows the results. It can be concluded that the gray entropy correlation degree of each evaluation index of the low-temperature viscoelastic properties from large to small are $T_C > m/S > m > S > W_d(t)/W_s(t) > \lambda > \Delta T_C$. Therefore, when using rheological methods to evaluate the PMA performance in the low-temperature range, it is recommended that the low-temperature continuous grading temperature $T_C$ and $m/S$ be preferred, because both $T_C$ and $m/S$ take into account the modulus and relaxation capacity of PMA. Where there is a contradiction between using single $S$ or $m$ to evaluate PMA's low-temperature performance, $m$ should be preferred. The dissipated energy ratio and relaxation time calculated by the Burgers model can reflect the PMA's cold behavior. Compared to the relaxation time, the dissipated energy ratio is more relevant to evaluate the low-temperature performance of PMA. The smallest correlation degree of gray entropy is between $\Delta T_C$ and the $\varepsilon_B$ in the low-temperature bending test. It is shown that although $\Delta T_C$ is not adequate to evaluate the low-temperature properties of PMA, it can indicate whether the PMA is $S$ or $m$ controlled.

**Table 4.** Gray entropy correlation.

| $E(S)$ | $E(m)$ | $E(T_C)$ | $E(\Delta T_C)$ | $E(m/S)$ | $E(\lambda)$ | $E(W_d(t)/W_s(t))$ |
|--------|--------|----------|-----------------|----------|--------------|---------------------|
| 0.9957 | 0.9982 | 0.9997 | 0.9329 | 0.9996 | 0.9592 | 0.9838 |

## 4. Conclusions

This study selected PMA (SBS-modified asphalt, rubber-modified asphalt, and high-viscosity-modified asphalt) and seven evaluation indexes of low-temperature viscoelastic properties. It used gray relational analysis to clarify the key indexes for evaluating the viscoelastic properties of PMA at low temperatures.

(1) The study used $S$, $m$, $T_C$, $\Delta T_C$, $m/S$, $\lambda$ as well as $W_d(t)/W_s(t)$ as to evaluate PMA's viscoelastic properties at low temperature. Rubber-modified asphalt exhibits better viscoelastic properties at low temperature and high-viscosity-modified asphalt than SBS-modified asphalt. In addition, PAM formed from rubber-modified asphalt has better cracking resistance at low temperature, which can confirm that rubber asphalt has a better application prospect at low temperature.

(2) The maximum bending strain of the PAM low-temperature bending experiment was selected as the reference sequence, and the $S$, $m$, $T_C$, $\Delta T_C$, $m/S$, $\lambda$ and $W_d(t)/W_s(t)$ were taken as the comparison sequence following the gray relational analysis. The order of the gray entropy correlation degree from large to small was $T_C > m/S > m > S > W_d(t)/W_s(t) > \lambda > \Delta T_C$. It is suggested that $T_C$ and $m/S$ should be preferred when applying the rheological low-temperature performance evaluation method of PMA. When only $S$ or $m$ as the evaluation index is contradictory, $m$ should be selected. Choosing the right evaluation index can make road construction more scientific and standard.

(3) $\Delta T_C$ is not recommended for evaluating the polymer-modified asphalt's viscoelastic properties. Still, it determines whether the PMA's low-temperature performance is dominated by $S$ or $m$, which should be paid attention to in future research.

(4) This study will have positive guiding significance for the better application of polymer-modified asphalt pavement in the future, especially when paving in low-temperature areas.

(5) This study about the low-temperature property of polymer-modified asphalt was limited to laboratory tests. In the further step, we will start from the engineering practice and combine the research results of laboratory tests with the asphalt selection and the actual situation of low-temperature cracking in cold areas.

**Author Contributions:** Conceptualization, Z.H.; methodology, Z.W.; data curation, J.L.; validation, P.L. and D.W.; investigation, Z.F. and W.Z.; resources, B.W.; writing—original draft preparation, R.L.; project administration, X.W. All authors have read and agreed to the published version of the manuscript.

**Funding:** This research was funded by the Scientific and Technological Development Project of Guangxi Communications Investment Group Corporation Ltd., grant number 2021-001.

**Institutional Review Board Statement:** Not applicable.

**Informed Consent Statement:** Informed consent was obtained from all subjects involved in the study.

**Data Availability Statement:** All data included in this study are privacy.

**Conflicts of Interest:** The authors declare no conflict of interest.

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
