# Peer review of "Investigation of Viscoelastic Properties of Polymer-Modified Asphalt at Low Temperature Based on Gray Relational Analysis"

_sustainability, doi:10.3390/su15086858_

Round 1

Reviewer 1 Report

1.       “the main diseases of polymer-modified asphalt” ‘Disease’ may not be an appropriate word. It should be replaced.

2.       Line 116, Obsjective should be replaced by Objective

3.       Line 123, recommend may be replaced by recommended or made recommendation like……

4.       Table 1, softening point of SBS 94.2, HVMA-II 85.9. But the dynamic viscosity SBS lower than Viscosity of HVMA-II, why

5.       BBR is the only test performed here? Author may include another test which is suitable

6.       What is the specification for CONTROL binder. So that it may be compared with SBS, HVMA……..

7.       What is the output of Burger model, did not understand

8.       Section 3.6 need to be elaborated

9.       Why author adopted Grey Relational Analysis, significance???

1.    PG grade of all the modified binder, specify

1.    Whole manuscript need to be improved

h.     More experimental test and results required

.f.     Manuscript concentrate on limited study area only

Reviewer 2 Report

SUMMARY

The article submitted for review is relevant. The authors evaluated the viscoelastic properties of polymer-modified asphalt at low temperatures based on Grey relational analysis. The scientific problem of the study is that the indicators for evaluating the low-temperature viscoelastic properties of polymer-modified asphalt PMA are currently not clear. Therefore, the authors investigated the stiffness, creep, creep rate, continuous classification temperature at low temperatures, carried out sequential comparison, used deformation analysis, as well as Grey relational analysis to evaluate the low-temperature viscoelastic properties of the resulting PMA. As a result of their research, the authors obtained important conclusions that are recommended for the construction of asphalt pavements in areas with low temperatures. All this leads to savings in time and money. Thus, the authors' study is interesting and useful. At the same time, the reviewer had a number of serious remarks. These comments are presented below.

COMMENTS

1.        The authors use the term "Evaluation" in their title. Probably, the authors should introduce the term “Investigation of viscoelastic properties…” in order to emphasize the research nature of the article. The practical value of the article is obvious, but there is not enough scientific novelty in it, and the reviewer suggests the authors to start with the title.

2.        The abstract also has flaws, in particular at the end of the abstract. The authors report that their result is a better orientation in the construction of asphalt pavement in areas with low temperatures. The authors also report time and cost savings, but do not quantify these results. I would like to understand, at least in approximate order, how much labor and material costs are reduced. This should be reflected in the abstract.

3.        The literature review conducted by the authors is insufficient to express from this the specific goal, tasks and scientific novelty of the study. The authors analyzed 24 references, while many of them are mentioned, but their essence of the study is not reflected. I would like to understand more clearly and in detail the purpose of the analysis, from which 2 problems will follow: an applied problem and a scientific problem, that is, a scientific deficit. The authors should improve the Introduction section with clear statements of the purpose, scientific novelty, practical significance of the study, and probably more references should be analyzed, since modified asphalts have recently been the subject of many scientific studies around the world.

4.        In the Obsjective section, that is, the objective, the authors are encouraged to formulate not only the objective, but also the tasks of the study. In addition, this section probably does not need to be separated into a independent section. Perhaps you should combine the sections "Introduction" and "Obsjective". There is a typo in the title of section 1 that needs to be corrected.

5.        I would like to see not just a listing of the selected materials in subsection 2.1 and methods in subsection 2.2, but their detailed justification, why these particular materials were chosen and what the authors based on. That is, the so-called analytical part is missing. I would like to see more reasonable work.

6.        The “Results and discussions” section contains many interesting results, however, the graphical dependencies in the form of histograms, firstly, are presented in not very high quality, and secondly, they are not described in sufficient detail - this applies to figures 1, 2 and 3. I wanted I would like to see a more detailed explanation of the obtained dependencies.

7.        Figure 6 is also interesting, but it probably does not contain enough data to talk about the exact nature of the relationship.

8.        Three temperatures are presented ‒24 °Ð¡, ‒18 °Ð¡, ‒12 °Ð¡, and only approximate dependences are given, connected between points by straight lines. However, the sections between these points also look interesting. Perhaps the authors did not quite correctly present the type of graphical dependence. They are encouraged to use, probably, a different kind of graphical dependence, and not a graph connected by straight lines.

9.        The same remark applies to figures 8 and 9. I would like to see more analysis and more comparisons of the results obtained with the results of other authors, then the discussion will clearly emphasize the scientific novelty of the study and characterize the achievement of the objective.

10.     A separate section called "Grey Relational Analysis" should probably also be included in the results and discussion. Authors are encouraged to consider how Sections 3 and 4 can be merged.

11.     It is recommended to supplement the conclusions in terms of a clear formulation of the scientific result, information about whether new knowledge was obtained or existing ideas developed, as well as a reflection of the vectors for continuing this research and the prospects for the practical application of these results for specific types of construction and objects.

12.     The list of references includes 34 points, but, as mentioned above, much more research is devoted to the topic of modified asphalts. The authors are recommended to increase the number of analyzed literature to at least 50 references, then the scientific novelty will become more obvious.

13.     The authors have worked out the topic of the analytical component well, but I would also like to see a certain number of graphic images, perhaps the article lacks photographs of the structure or a visual image of the material being studied. The author is recommended to work on the attractiveness of the article so that it becomes more interesting to the reader.

14.     The general remark on the article is as follows. The article is interesting, relevant, the chosen topic is acutely on the agenda in science and engineering practice. At the same time, the article has a number of shortcomings that the reviewer listed, and they need to be corrected. The article needs serious improvements and after all the improvements it must be re-sent for review.

Author Response

Thanks a lot for your comments. Please see the attachment.

Reviewer 3 Report

This research aims to evaluate different methods for evaluating performance at low temperatures. The research conducted is interesting. However, in some parts, the topic could be better explained. Attached is the file with my comments.

Reviewer 4 Report

This research compared the results of 6 low temperature indexes in 4 types of modified binders with the corresponding porous mixture to find the best index of low temperature cracking evaluation. BBR and low temperature bending tests were used for binders and mixtures, respectively. The research is good for publication but requires some improvement. Please find my comments below:  

English needs to improve significantly. There are typos such as delta Tc in section 3.3.

The abstract is unclear to follow. What is the novelty? Delta Tc is already there and it is well-established that the low temperature PG of the almost all binders is determined by m-value, which is called Tc. what is the novelty the manuscript?  

There must be a flowchart indicating the testing program in a section called methodology. Within methodology test must be explained. The low-temperature bending needs to be explained there and also the indexes. Now, it is not well-organized and methodology and results are all together.

It is also highly recommended to show some pics of the Lab work done.

The authors needs to justify the use of porous asphalt. Even in one simple sentence.

Round 2

Reviewer 1 Report

Author has answered all the comments raised by the reviewer. The paper may be accepted for publication.

Reviewer 2 Report

The authors have significantly improved the manuscript by responding to most of the reviewer's comments. The article is now ready for publication.